# ChronoEdit: Towards Temporal Reasoning for Image Editing and World Simulation

**Jay Zhangjie Wu**[1,*]  **Xuanchi Ren**[1,2,*]  **Tianchang Shen**[1,2]  **Tianshi Cao**[1,2]  **Kai He**[1,2]
**Yifan Lu**[1]  **Ruiyuan Gao**[1]  **Enze Xie**[1]  **Shiyi Lan**[1]  **Jose M. Alvarez**[1]  **Jun Gao**[1]
**Sanja Fidler**[1,2]  **Zian Wang**[1,2]  **Huan Ling**[1,*,†]
[1]NVIDIA  [2]University of Toronto

## Abstract

Recent advances in large generative models have greatly enhanced both image editing and in-context image generation, yet a critical gap remains in ensuring physical consistency, where edited objects must remain coherent. This capability is especially vital for world simulation related tasks. In this paper, we present ChronoEdit, a framework that reframes image editing as a video generation problem. First, ChronoEdit treats the input and edited images as the first and last frames of a video, allowing it to leverage large pretrained video generative models that capture not only object appearance but also the implicit physics of motion and interaction through learned temporal consistency. Second, ChronoEdit introduces a temporal reasoning stage that explicitly performs editing at inference time. Under this setting, target frame is jointly denoised with reasoning tokens to imagine a plausible editing trajectory that constrains the solution space to physically viable transformations. The reasoning tokens are then dropped after a few steps to avoid the high computational cost of rendering a full video. To validate ChronoEdit, we introduce PBench-Edit, a new benchmark of image–prompt pairs for contexts that require physical consistency, and demonstrate that ChronoEdit surpasses state-of-the-art baselines in both visual fidelity and physical plausibility. Project page for code and models: https://research.nvidia.com/labs/toronto-ai/chronoedit

## 1 Introduction

Recent large-scale generative models have transformed the landscape of image editing, enabling purely text-driven image modifications that impact diverse domains such as social media, e-commerce, education, and creative arts (Xiao et al., 2025; Labs et al., 2025; Liu et al., 2025; Yu et al., 2025). Beyond these consumer applications, image editing also offers a critical capability for simulation-related applications, providing a controllable mechanism to simulate rare but safety-critical scenarios that are otherwise difficult to capture in real-world data. For example, editing can create long-tail events for autonomous driving, where unexpected objects enter the road (Gao et al., 2024; Lu et al., 2024), or visualize the outcomes of a robot arm manipulating objects in a cluttered scene. In these cases, editing goes beyond aesthetics and serves to generate diverse training and evaluation data for perception, planning, and reasoning.

A central requirement in the context of image editing for simulation is *physical consistency*: edited results must preserve existing objects and their properties (*e.g.*, color, geometry) while reflecting the intended change. Without this constraint, edited results risk misrepresenting the scene and compromising downstream systems. Existing editing models have explored character or object continuity using video data to curate pixel-level editing pairs (Deng et al., 2025; Xiao et al., 2025; Chen et al., 2025), yet data alone have failed to ensure physical consistency. As illustrated in Fig. 2, these models often hallucinate new objects or alter geometry in unintended ways. Such failures stem partly from architectural limitations: current approaches are purely data-driven and lack mechanisms

---

*Equal contribution; † Corresponding author

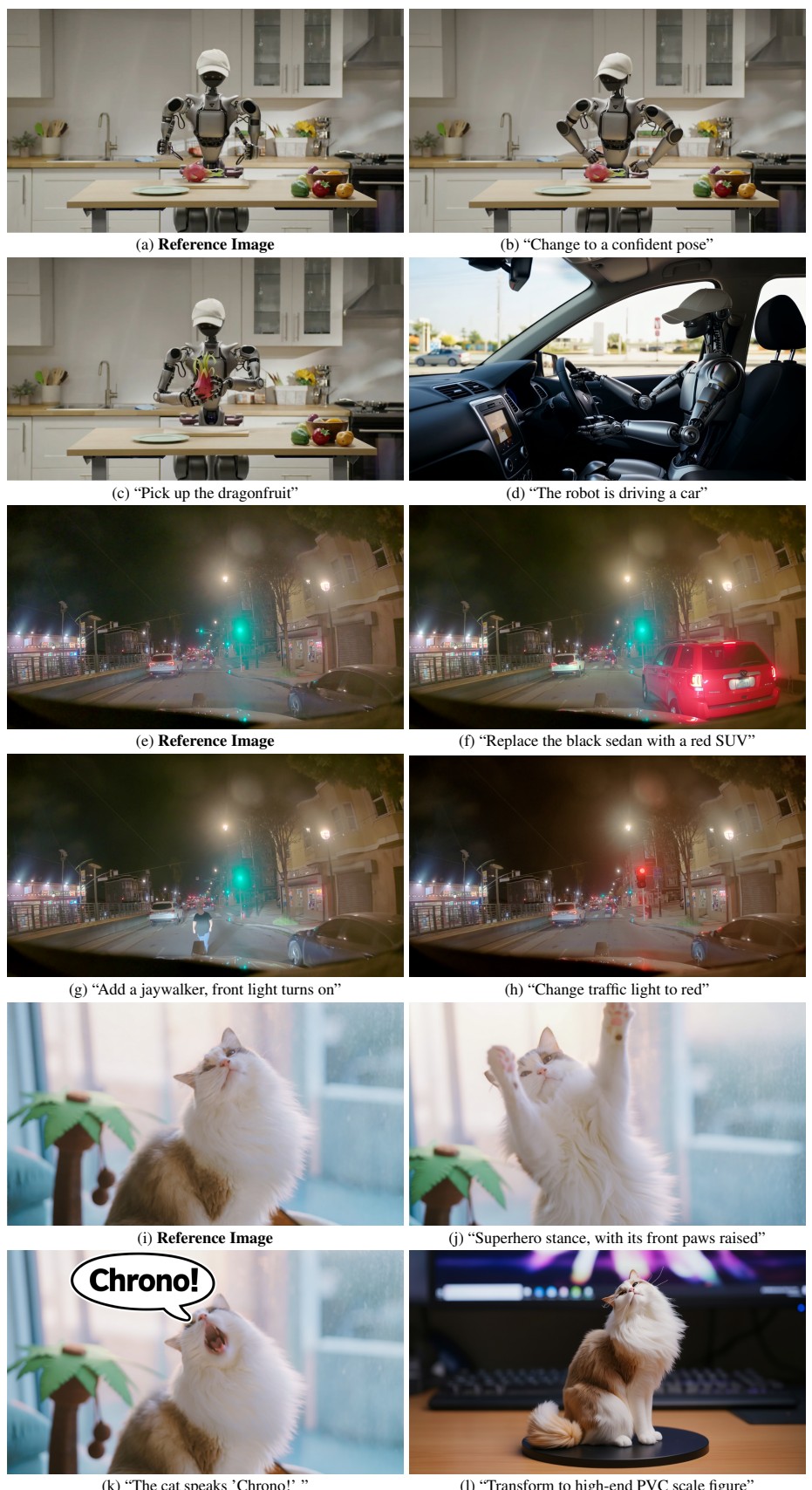

(a) **Reference Image**

(b) "Change to a confident pose"

(c) "Pick up the dragonfruit"

(d) "The robot is driving a car"

(e) **Reference Image**

(f) "Replace the black sedan with a red SUV"

(g) "Add a jaywalker, front light turns on"

(h) "Change traffic light to red"

(i) **Reference Image**

(j) "Superhero stance, with its front paws raised"

(k) "The cat speaks 'Chrono!' "

(l) "Transform to high-end PVC scale figure"

Figure 1: **Physical consistent image editing results with ChronoEdit-14B.** ChronoEdit produces edits that are both visually convincing and physically consistent with the underlying scene context.

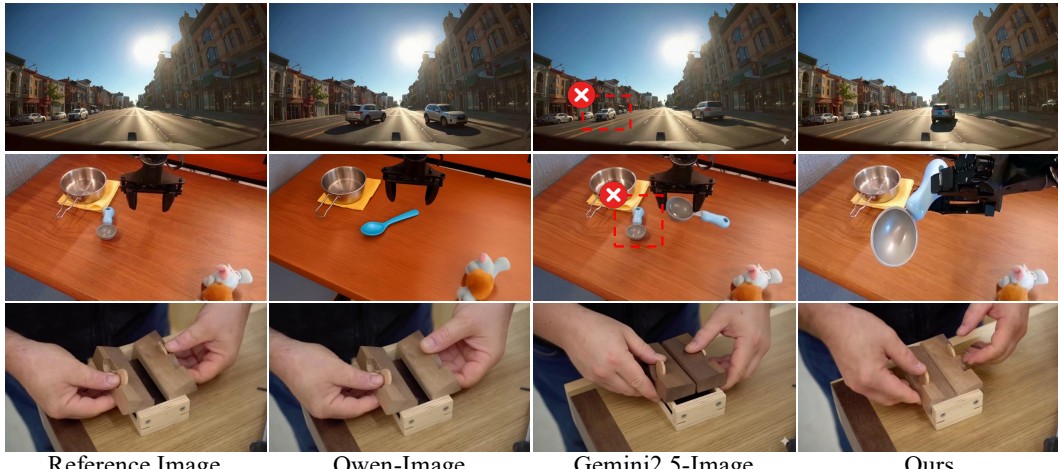

|                 |            |                |      |
| Reference Image | Qwen-Image | Gemini2.5-Image | Ours |

Figure 2: **Failure cases of state-of-the-art image editing models.** Current state-of-the-art models often struggle to maintain physical consistency on world simulation-related editing tasks. They may hallucinate unintended objects or distort scene geometry. In contrast, our method produces edits that are faithful to the instruction and remain coherent with the scene. Prompts (from top to bottom): (1) "The left silver SUV makes a U-turn", (2) "Pick up the spoon with the robot arm", and (3) "Close the wooden piece by hand".

to enforce continuity, leaving them vulnerable to drifting edits that appear plausible but violate physical constraints.

Large-scale video generative models (Wan, 2025; Cosmos, 2025) have recently demonstrated strong capabilities to preserve object structure and coherence across consecutive frames. This inherent temporal prior makes them particularly well-suited for editing tasks that demand physical consistency. Building upon this insight, we introduce ChronoEdit, a foundation model for image editing explicitly designed to preserve physical consistency. ChronoEdit repurposes pretrained video generative models for editing by reframing the task as a *two-frame* video generation problem, where the input image and its edited version are modeled as consecutive frames. When fine-tuned with curated image-editing data, this two-frame formulation equips the video model with editing functionalities while leveraging its pretrained temporal prior to preserve object fidelity.

For world simulation tasks (e.g., action editing) that demand stronger temporal coherence, we further introduce explicitly guided editing through temporal reasoning. Given an input image and an editing instruction, the model imagines and denoises a short video trajectory that realizes the edit while preserving temporal alignment with the input frame. The intermediate video frames in this trajectory act as reasoning tokens, planning how the edit should unfold and thereby producing more physically plausible results (See Fig. 2). Beyond improving plausibility, simulating these intermediate frames also unveils the "thinking process" of the editing model, offering a more interpretable view of how edits are constructed. To balance these benefits with efficiency, ChronoEdit can perform temporal reasoning during only the first few high-noise denoising steps. After that stage, the intermediate frames are discarded, and only the final frame of the trajectory is refined into the edited image.

Public benchmarks for image editing mainly target visual fidelity and instruction following, but rarely evaluate physical consistency. To address this gap, we introduce a new benchmark named PBench-Edit. This benchmark is constructed by carefully curating a collection of images paired with editing prompts that capture both real-world editing requirements and a broad spectrum of editing types. PBench-Edit is designed to evaluate not only general-purpose edits but also tasks that require physical and temporal consistency. Our experiments on PBench-Edit demonstrate that ChronoEdit achieves state-of-the-art results, surpassing existing open-source baselines by a significant margin and narrowing the gap with leading proprietary systems.

In summary, we make the following contributions: **(i)** We propose ChronoEdit, a foundation model for image editing designed to preserve physical consistency. **(ii)** We present an effective design that turns a pretrained video generative model into an image editing model. **(iii)** We develop a novel temporal reasoning inference stage that further enforces physical consistency. **(iv)** We demonstrate that ChronoEdit achieves state-of-the-art performance among open-source models and is competitive with leading proprietary systems. **(v)** We present a benchmark tailored to world simulation applications.

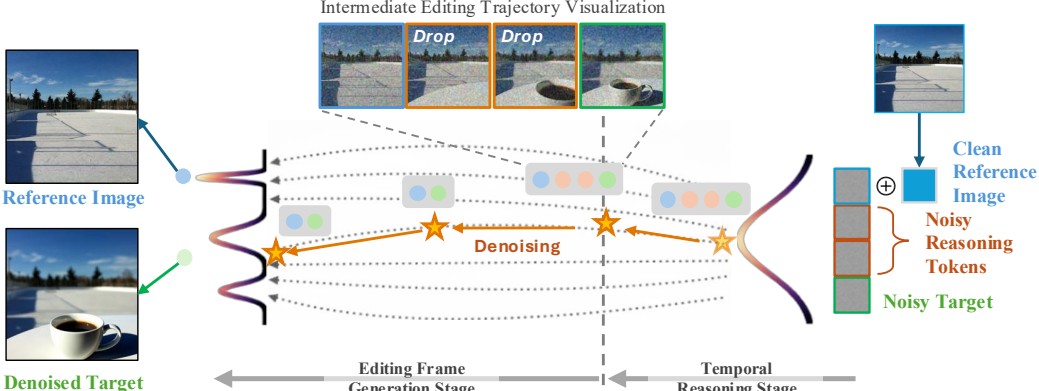

Figure 3: **Overview of the ChronoEdit pipeline.** From right to left, the denoising process begins in the *temporal reasoning stage*, where the model imagines and denoises a short trajectory of intermediate frames. These intermediate frames act as reasoning tokens, guiding how the edit should unfold in a physically consistent manner. For efficiency, the reasoning tokens are discarded in the subsequent *editing frame generation stage*, where the target frame is further refined into the final edited image.

## 2 RELATED WORK

Recent advances in image editing have been driven by large-scale foundation models. FLUX.1 Kontext (Labs et al., 2025) achieves strong instruction alignment and multi-turn editing through billion-scale parameterization, while OmniGen (Xiao et al., 2025) unifies text-to-image, editing, and subject-driven generation within a single diffusion framework. Qwen-Image-Edit (Wu et al., 2025a) extends a vision-language model with a double-stream architecture for precise, high-fidelity edits. Proprietary systems such as GPT-4o (OpenAI, 2025) and Gemini 2.5 Flash Image (Google, 2025) demonstrate robust multi-turn editing and conversational refinement at scale.

Prior work also explored leveraging video priors for image editing: Bagel (Deng et al., 2025), UniReal (Chen et al., 2025), and OmniGen (Xiao et al., 2025) use video-derived key frames to create temporally coherent image pairs. In a complementary direction, Rotstein et al. (2025) is a training-free method that uses a pretrained image-to-video diffusion model to synthesize a sequence of intermediate frames, and then selects the frame that best satisfies the edit. Concurrent work (Wiedemer et al., 2025) shows that strong video models like Veo 3 better preserve fine details and textures during edits.

*A complete discussion of related work can be found in Appendix A.*

## 3 CHRONOEDIT

In this section, we first provide background on the rectified flow model for video generation in Sec. 3.1. Next, we outline our core design in Sec. 3.2, which adapts a pretrained image-to-video model for image editing, and detail our training with video reasoning tokens. We then describe the inference procedure in Sec. 3.3, highlighting how video reasoning enhances consistency. Finally, Sec. 3.4 describes the post-training process of ChronoEdit, which improves inference speed. An overview of the full architecture is shown in Fig. 3.

### 3.1 BACKGROUND: RECTIFIED FLOW FOR VIDEO GENERATIVE MODELS

Modern video generative models typically rely on a pretrained variational autoencoder (VAE) (Blattmann et al., 2023b;a; Kong et al., 2024; Gupta et al., 2024; Cosmos, 2025; Wan, 2025) that compresses raw videos $\mathbf{x} \in \mathbb{R}^{F \times 3 \times H \times W}$ into a compact latent representation $\mathbf{z}_0 = \mathcal{E}(\mathbf{x}) \in \mathbb{R}^{F' \times C \times h \times w}$. Training and inference operate in this latent space, and the decoder reconstructs videos as $\hat{\mathbf{x}} = \mathcal{D}(\mathbf{z})$. To handle temporal structure, causal video VAEs encode the first frame independently and compress subsequent chunks conditioned on past latents. In our work, we adopt the Wan2.1 VAE (Wan, 2025), which yields $F' = \frac{(F-1)}{4} + 1$, $C = 16$, $h = H/8$, and $w = W/8$.

Rectified flow (Liu et al., 2022; Albergo & Vanden-Eijnden, 2022; Lipman et al., 2022; Esser et al., 2024a) provides a principled framework for training video generators via flow matching. Given

video data $\mathbf{x} \sim p_{\text{data}}$ and Gaussian noise $\epsilon \sim \mathcal{N}(\mathbf{0}, \boldsymbol{I})$, the interpolated latent at timestep $t \in [0, 1]$ is defined as $\mathbf{z}_t = (1 - t)\mathbf{z}_0 + t\epsilon$, with $\mathbf{z}_0 = \mathcal{E}(\mathbf{x})$. A denoiser $\mathbf{F}_{\boldsymbol{\theta}}(\mathbf{z}_t, t; \mathbf{y}, \mathbf{c})$ with parameters $\boldsymbol{\theta}$ is trained to predict the target velocity field $(\epsilon - \mathbf{z}_0)$ by minimizing:

$$\mathcal{L}_{\boldsymbol{\theta}} = \mathbb{E}_{t \sim p(t),\, \mathbf{x} \sim p_{\text{data}},\, \epsilon \sim \mathcal{N}(\mathbf{0}, \boldsymbol{I})} \left[ \left\| \mathbf{F}_{\boldsymbol{\theta}}(\mathbf{z}_t, t; \mathbf{y}, \mathbf{c}) - (\epsilon - \mathbf{z}_0) \right\|_2^2 \right], \tag{1}$$

Here $\mathbf{y}$ denotes optional text conditioning and $\mathbf{c}$ is optional image conditioning.

## 3.2 Re-purposing Video Generative Models for Editing

Formally, the image editing task aims to transform a reference image $\mathbf{c}$ into an output image $\mathbf{p}$ that satisfies a natural-language instruction $\mathbf{y}$. Our key insight is to repurpose a pretrained image-to-video model for this task, leveraging its inherent temporal priors to maintain consistency between the source and target images.

**Encoding Editing Pairs.** To leverage temporal priors in pretrained video models, we reinterpret the editing pair $\{\mathbf{c}, \mathbf{p}\}$ as a short video sequence. Specifically, the input image is encoded as the first latent frame $\mathbf{z}_{\mathbf{c}} = \mathcal{E}(\mathbf{c})$, while the output image is repeated four times to match the video VAE's $4\times$ temporal compression and encoded as $\mathbf{z}_{\mathbf{p}} = \mathcal{E}(\texttt{repeat}(\mathbf{p}, 4))$. This produces two temporal latents aligned with the video model's architecture. We additionally adjust the model's 3D-factorized Rotary Position Embedding (RoPE) (Su et al., 2024) by anchoring input image $\mathbf{c}$ at timestep $0$ and output image $\mathbf{p}$ at a predefined timestep $T$, explicitly encoding their temporal separation. For convenience, we fix $T$ to the length of the joint-training video latents (see following section).

**Temporal Reasoning Tokens.** To go beyond direct input–output mapping, we explicitly model the transition between input image $\mathbf{c}$ and output image $\mathbf{p}$. The goal is to encourage the model to imagine a plausible trajectory rather than regenerate the target image in a single step, which often leads to abrupt changes. By reasoning through intermediate states, the model better preserves object identity, geometry, and physical coherence. In practice, we insert intermediate latent frames between $\mathbf{z}_{\mathbf{c}}$ and $\mathbf{z}_{\mathbf{p}}$. These frames are initialized with random noise and denoised jointly with the output frame latents. We refer to them as temporal *reasoning tokens* $\mathbf{r}$, since they act as intermediate guidance that help the model "think" through plausible transitions.

**Unifying Image Pairs and Videos.** Similarly to the video denoiser introduced in Sec. 3.1, we define the image-editing denoiser as $\mathbf{F}_{\boldsymbol{\theta}}(\mathbf{z}_{\mathbf{p},t}, t; \mathbf{y}, \mathbf{z}_{\mathbf{c}})$, where $\mathbf{z}_{\mathbf{p},t}$ and $t$ are the flow variables. Our formulation naturally supports training on both image-editing pairs and full video sequences within a unified framework. For public image-editing datasets, each pair $(\mathbf{c}, \mathbf{p}, \mathbf{y})$ is reinterpreted as a two-frame video, where $\mathbf{c}$ is the first frame and $\mathbf{p}$ the last, directly supervising instruction-based edits. For videos, the structure matches our reasoning-token design: the first frame corresponds to $\mathbf{c}$, the last corresponds to $\mathbf{p}$, and all intermediate frames serve as reasoning tokens. Input and reasoning frames are encoded into latents by the video VAE as standard video frames, while the target frame is separately encoded and repeated four times to match the VAE's temporal compression. This design makes reasoning tokens optional at inference—the VAE decoder can still recover the target frame independently—while providing strong supervision for coherent transitions when present. Together, this joint training strategy allows the model to learn semantic alignment from image pairs while additionally learn temporal consistency grounded in video data.

**Video Data Curation.** Training with reasoning tokens requires diverse examples of how scenes evolve over time. To this end, we curate a large-scale synthetic dataset of 1.4M videos generated with state-of-the-art video generative models. We place particular emphasis on disentangling scene dynamics from camera motion, since unintended viewpoint shifts between the first and last frames could be misinterpreted as edits during training.

Our corpus covers three complementary categories: (i) *Static-camera, dynamic-object* clips produced by text-to-video models (Wan, 2025; Cosmos, 2025), where we append the postfix "The camera remains stationary throughout the video." to prompts and filter unstable clips using ViPE (Huang et al., 2025); (ii) *Egocentric driving scenes*, a critical world-simulation scenario, generated with the HDMap-conditioned model of Ren et al. (2025a), which fixes the camera while explicitly controlling vehicle motion via bounding boxes; and (iii) *Dynamic-camera, static-scene* clips from GEN3C (Ren et al., 2025b), which allow precise camera trajectory control while keeping the scene content fixed. Finally, to provide corresponding instructions $\mathbf{y}$, we employ a VLM to caption each video with an editing instruction, summarizing the transition from input to output frame, as detailed in Appendix D.

## 3.3 Inference with Temporal Reasoning

To perform image editing efficiently at inference time, we introduce a two-stage method which allows the model to benefit from video reasoning tokens without incurring the full computational cost of generating a complete video. Intuitively, the first few noisiest steps of a flow/diffusion trajectory determine the global structure of the outcome, and hence tokens more frequently attend across frames in the sequence. Hence, we incorporate video reasoning tokens in these first denoising steps, and omit them in later denoising steps to obtain the best balance between quality and computational cost. Pseudocode is provided in Algo. 1 and visualization is shown in Fig. 3.

In the first stage, we concatenate clean input tokens $\mathbf{z_c}$, sampled reasoning tokens $\mathbf{r}$ and noisy sampled output tokens $\mathbf{z_p}$ into one temporal sequence. Similar to image-to-video generation, the model performs denoising on the concatenated sequence without modifying the $\mathbf{z_c}$ tokens. Rather than denoising all the way to clean latents, $N_r$ steps of denoising are performed, and the partially denoised last

---

**Algorithm 1** Sampling process of ChronoEdit. When Temporal Reasoning is enabled, $N_r$ steps are taken with video reasoning tokens $\mathbf{z}_{full}$. Setting $r = 0$ or $N_r = 0$ recovers standard sampling w/o Temporal Reasoning.

---

**Given:** Denoising model $\mathbf{F}_\theta$, ODE solver $\texttt{ODESolve}(v_t, t, \mathbf{z}_t, t') \rightarrow \mathbf{z}_{t'}$, trajectory reasoning length $r$, sequence of $N$ time steps $T = (t_{max}, \dots, t_{min})$, reasoning timestep $N_r$.
**Input:** Editing instruction tokens $\mathbf{y}$, input image token $\mathbf{c}$
   $\epsilon \sim \mathcal{N}(0, I)$ with shape $(r + 1) \times C \times h \times w$
   $\mathbf{z}_{full} \leftarrow \texttt{concat}(\mathbf{c}, \epsilon)$
   $n \leftarrow 0, t \leftarrow T[0]$
   **while** $n < N$ **do**
     **if** $n < N_r$ **then**
       $\mathbf{v} \leftarrow \mathbf{F}_\theta(\mathbf{z}_{full}, t; \mathbf{y}, \mathbf{c})$
       $\mathbf{z}_{full} \leftarrow \texttt{ODESolve}(\mathbf{v}, t, \mathbf{z}_{full}, T[n+1])$
     **else if** $n \geq N_r$ **then**
       **if** $n = N_r$ **then**
         $\mathbf{z}_{final} \leftarrow \texttt{concat}(\mathbf{c}, \mathbf{z}_{full}[-1])$
       $\mathbf{v} \leftarrow \mathbf{F}_\theta(\mathbf{z}_{final}, t; \mathbf{y}, \mathbf{c})$
       $\mathbf{z}_{final} \leftarrow \texttt{ODESolve}(\mathbf{v}, t, \mathbf{z}_{final}, T[n+1])$
     $n \leftarrow n + 1, t \leftarrow T[n]$
   $\mathbf{x} \leftarrow \texttt{Decode}(\mathbf{z}_{final})[-1]$
**Output:** $\mathbf{x}$

---

latents of the temporal sequence corresponding to $\mathbf{z_p}$ are carried forward. In the second stage, the partially denoised output latent is concatenated behind the clean input latent and fully denoised for the remaining $N - N_r$ steps. As in training, the output latent corresponds to four repeated frames to match the video VAE's temporal compression. After decoding to RGB, the four frames typically collapse to the same image, and we take the last frame as the final edited result.

## 3.4 Few-Step Distillation for Fast Inference

To further accelerate inference, we employed distillation techniques to reduce the number of steps required for inference. Specifically, we utilized DMD loss (Yin et al., 2024) to train an 8-step student model. The gradient of the distillation objective is given by

$$\nabla \mathcal{L}_{\text{DMD}} = -\mathbb{E}_t \left( \int \left( s_{\text{real}}(f(\mathbf{F}_\theta, t), t) - s_{\text{fake}}(f(\mathbf{F}_\theta, t), t) \frac{d\mathbf{F}_\theta}{d\theta} dz \right) \right), \tag{2}$$

where $s_{\text{real}}$ and $s_{\text{fake}}$ denote the score estimation from the teacher model and a trainable fake score model, respectively; $f(\cdot)$ is the forward diffusion process (*i.e.*, noise injection). We omit the conditioning term for simplicity. Through this training process, our model can significantly improve the inference speed while maintaining prompt-following ability and image editing quality.

## 4 Experiments

We evaluate ChronoEdit in two configurations, with 14B and 2B parameters, denoted as ChronoEdit-14B and ChronoEdit-2B. We evaluate both models across multiple datasets and editing tasks, compare them with both open-source and proprietary baselines, and ablate the contribution of different design choices. We further evaluate the variant of ChronoEdit-14B with temporal reasoning (*ChronoEdit-14B-Think*), and the step distillation (*ChronoEdit-14B-Turbo*).

**Training Details.** ChronoEdit-14B is finetuned from the pretrain model of Wan2.1-I2V-14B-720P[1] (Wan, 2025) and ChronoEdit-2B is built upon Cosmos-Predict2.5-2B[2] (Cosmos, 2025). Both

---

[1]https://huggingface.co/Wan-AI/Wan2.1-I2V-14B-720P
[2]https://research.nvidia.com/labs/dir/cosmos-predict2.5

| Model | Model Size | Add | Adjust | Extract | Replace | Remove | Background | Style | Hybrid | Action | Overall ↑ |
|---|---|---|---|---|---|---|---|---|---|---|---|
| MagicBrush (Zhang et al., 2023a) | 0.9B | 2.84 | 1.58 | 1.51 | 1.97 | 1.58 | 1.75 | 2.38 | 1.62 | 1.22 | 1.90 |
| Instruct-Pix2Pix (Brooks et al., 2023) | 0.9B | 2.45 | 1.83 | 1.44 | 2.01 | 1.50 | 1.44 | 3.55 | 1.20 | 1.46 | 1.88 |
| AnyEdit (Yu et al., 2025) | 0.9B | 3.18 | 2.95 | 1.88 | 2.47 | 2.23 | 2.24 | 2.85 | 1.56 | 2.65 | 2.45 |
| UltraEdit (Zhao et al., 2024) | 8B | 3.44 | 2.81 | 2.13 | 2.96 | 1.45 | 2.83 | 3.76 | 1.91 | 2.98 | 2.70 |
| OmniGen (Xiao et al., 2025) | 3.8B | 3.47 | 3.04 | 1.71 | 2.94 | 2.43 | 3.21 | 4.19 | 2.24 | 3.38 | 2.96 |
| ICEdit (Zhang et al., 2025) | 12B | 3.58 | 3.39 | 1.73 | 3.15 | 2.93 | 3.08 | 3.84 | 2.04 | 3.68 | 3.05 |
| Step1X-Edit (Liu et al., 2025) | 19B | 3.88 | 3.14 | 1.76 | 3.40 | 2.41 | 3.16 | 4.63 | 2.64 | 2.52 | 3.06 |
| BAGEL (Deng et al., 2025) | 7B-MoT | 3.56 | 3.31 | 1.70 | 3.3 | 2.62 | 3.24 | 4.49 | 2.38 | 4.17 | 3.20 |
| UniWorld-V1 (Lin et al., 2025) | 12B | 3.82 | 3.64 | 2.27 | 3.47 | 3.24 | 2.99 | 4.21 | 2.96 | 2.74 | 3.26 |
| OmniGen2 (Wu et al., 2025b) | 7B | 3.57 | 3.06 | 1.77 | 3.74 | 3.20 | 3.57 | 4.81 | 2.52 | 4.68 | 3.44 |
| FLUX.1 Kontext [Dev] (Labs et al., 2025) | 12B | 3.76 | 3.45 | 2.15 | 3.98 | 2.94 | 3.78 | 4.38 | 2.96 | 4.26 | 3.52 |
| FLUX.1 Kontext [Pro] (Labs et al., 2025) | N/A | 4.25 | 4.15 | 2.35 | 4.56 | 3.57 | 4.26 | 4.57 | 3.68 | 4.63 | 4.00 |
| GPT Image 1 [High] (OpenAI, 2025) | N/A | 4.61 | 4.33 | 2.90 | 4.35 | 3.66 | 4.57 | 4.93 | 3.96 | 4.89 | 4.20 |
| Qwen-Image (Wu et al., 2025a) | 20B | 4.38 | 4.16 | 3.43 | 4.66 | 4.14 | 4.38 | 4.81 | 3.82 | 4.69 | 4.27 |
| **ChronoEdit-2B** | 2B | 4.30 | 4.29 | 2.87 | 4.23 | 4.50 | 4.40 | 4.60 | 3.20 | 4.81 | 4.13 |
| **ChronoEdit-14B-Turbo (8 steps)** | 14B | 4.36 | 4.38 | 3.28 | 4.11 | 4.00 | 4.31 | 4.31 | 3.67 | 4.78 | 4.13 |
| **ChronoEdit-14B** | 14B | 4.48 | 4.39 | 3.49 | 4.66 | 4.57 | 4.67 | 4.83 | 3.82 | 4.91 | 4.42 |

Table 1: Quantitative comparison results on ImgEdit (Ye et al., 2025). All metrics are evaluated by GPT-4.1. "Overall" is calculated by averaging all scores across tasks.

models are trained using a learning rate of $2e-5$ and weight decay of $1e-3$. Since the pretrained model already exhibits strong capability in generating fine-grained details, we sample timesteps $t \in [0, 1]$ from a logit-normal distribution with shift value set to 5 (Esser et al., 2024b), thereby oversampling the large-timestep region. The model is pretrained on 1.4 million videos and 2.6 million image pairs, with the first and last frames of each video also included as additional image pairs. During training, we adopt a 1:1 ratio between image pairs and videos, where the video data is used to learn video reasoning tokens. We empirically use 6 intermediate latent frames as temporal reasoning tokens, corresponding to 24 frames in pixel space, which results in a total of $T = 8$ timesteps. Training is performed with a batch size of 128. In the final stage, the pretrained model is fine-tuned on a high-quality supervised fine-tuning (SFT) dataset of 50k images and 20k videos, sampled at a 5:1 ratio for 10k steps. For ChronoEdit-14B-Turbo, we apply distillation loss with a learning rate of $2e-6$ for 1500 steps, setting the update ratio between the student and the fake score model (Yin et al., 2024) to 5 for stable training.

**Benchmarks.** We evaluate our method on two complementary benchmarks. First, for general-purpose image editing, we use the ImgEdit-Basic-Edit Suite (Ye et al., 2025) which consists of 734 test cases spanning nine common image-editing tasks: add, remove, alter, replace, style transfer, background change, motion change, hybrid edit, and action. The benchmark is constructed from manually collected Internet images to ensure semantic diversity, with the action category primarily emphasizing human pose modifications. Model performance on each task is evaluated using GPT-4.1, which measures adherence to instructions, quality of the edit, and detail preservation (Ye et al., 2025).

While prior benchmarks for image editing assess visual realism and instruction alignment, they provide limited evaluation of physical consistency. We therefore develop PBench-Edit, an image-editing benchmark derived from the original PBench dataset (Pbench, 2025), designed to assess editing in physically grounded contexts.

The original PBench evaluates world-model progress in domains such as autonomous driving, robotics, physics, and common-sense reasoning. PBench-Edit repurposes its curated videos and captions for targeted editing tasks by selecting representative frames from each domain and pairing them with manually verified editing instructions. Unlike ImgEdit-Action, PBench-Edit covers a broader spectrum of real-world interactions—such as cooking, driving, and robot manipulation—resulting in a benchmark that is both diverse and physically grounded. It includes 271 images in total (133 human, 98 robot, and 40 driving). Evaluation is performed with GPT-4.1 using the same criteria as ImgEdit (Ye et al., 2025): adherence to instructions, edit quality, and detail preservation. Additional visualizations are provided in Fig. S4.

## 4.1 QUANTITATIVE EVALUATION

**General-Purpose Image Editing Results.** Tab. 1 reports results on the ImgEdit Basic-Edit Suite (Ye et al., 2025). To ensure fair comparison with prior works in terms of compute cost, we disable Temporal Reasoning and evaluate ChronoEdit-14B as a pure image-editing model. ChronoEdit-14B achieves the highest overall score of 4.42, outperforming state-of-the-art baselines. Among open-source models, FLUX.1 Kontext [Dev] is the most comparable in scale (12B *vs.* 14B). ChronoEdit-14B surpasses it by +0.90 overall, with especially large improvements on extract (4.66 *vs.* 2.15, +2.51), remove (4.57 *vs.* 2.94, +1.63), while performing on par in style transfer (4.83 *vs.* 4.38). These results indicate the strong capability of ChronoEdit for instruction-driven edits that require spatial

| Model | Action Fidelity | Identity Preservation | Visual Coherence | Overall ↑ |
|---|---|---|---|---|
| Step1X-Edit (Liu et al., 2025) | 3.39 | 4.52 | 4.44 | 4.11 |
| BAGEL (Deng et al., 2025) | 3.83 | 4.60 | 4.53 | 4.32 |
| OmniGen2 (Wu et al., 2025b) | 2.65 | 4.02 | 4.02 | 3.56 |
| FLUX.1 Kontext [Dev] (Labs et al., 2025) | 2.88 | 4.29 | 4.32 | 3.83 |
| Qwen-Image (Wu et al., 2025a) | 3.76 | 4.54 | 4.48 | 4.26 |
| **ChronoEdit-14B** | 4.01 | **4.65** | 4.63 | 4.43 |
| **ChronoEdit-14B-Think** ($N_r = 10$) | **4.31** | 4.64 | **4.64** | **4.53** |
| **ChronoEdit-14B-Think** ($N_r = 20$) | 4.28 | 4.62 | 4.62 | 4.51 |
| **ChronoEdit-14B-Think** ($N_r = 50$) | 4.29 | 4.64 | 4.63 | 4.52 |
| **ChronoEdit-2B-Think** ($N_r = 10$) | 4.17 | 4.61 | 4.56 | 4.44 |

Table 2: Quantitative comparison results on PBench-Edit. All metrics are evaluated by GPT-4.1. "Overall" is calculated by averaging all scores across dimensions.

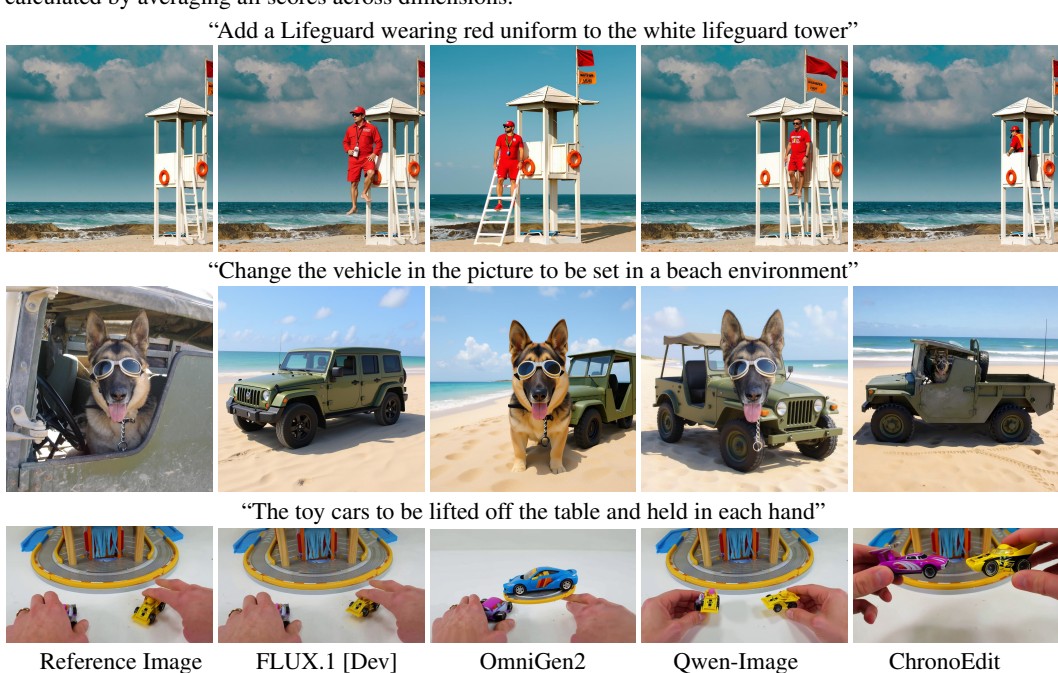

"Add a Lifeguard wearing red uniform to the white lifeguard tower"

"Change the vehicle in the picture to be set in a beach environment"

"The toy cars to be lifted off the table and held in each hand"

Reference Image      FLUX.1 [Dev]      OmniGen2      Qwen-Image      ChronoEdit

Figure 4: **Comparison with baseline methods.** The first two rows show examples from the ImageEdit Basic-Edit Suite (Ye et al., 2025) benchmark, and the last row is from PBench-Edit, where ChronoEdit-Think is evaluated with 10 temporal reasoning steps. In both benchmarks, ChronoEdit achieves edits that more faithfully follow the given instructions while preserving scene structure and fine details.

and structural reasoning. Compared to the 20B open-source model Qwen-Image (Wu et al., 2025a) which scores 4.27 overall, ChronoEdit-14B matches or outperforms its performance across all tasks. Notably, ChronoEdit-14B achieves stronger results on challenging categories such as background change (4.67 *vs.* 4.38) and action/motion edits (4.41 *vs.* 4.27), suggesting that joint image–video pretraining provides strong advantages for modeling dynamic consistency and scene transformations.

It is also worth noting that ChronoEdit-14B-Turbo, which runs 6× faster than ChronoEdit-14B (5.0s vs. 30.4s per image, with speeds measured on 2 Nvidia-H100 GPUs), achieves results only 0.3 points below ChronoEdit-14B, yet still outperforms FLUX.1 Kontext [Dev] and FLUX.1 Kontext [Pro] by margins of 0.61 and 0.13, respectively.

Moreover, we also report ChronoEdit-2B results, which are 7× smaller than ChronoEdit-14B but works on-par with ChronoEdit-14B-Turbo.

**World Simulation Editing Results.** We evaluate our method on the PBench-Edit benchmark, which emphasizes physically grounded editing scenarios. As shown in Tab. 2, ChronoEdit-14B achieves the highest overall score (4.43), outperforming strong baselines such as BAGEL (4.32), Qwen-Image (4.26), and FLUX.1 Kontext [Dev] (3.83). Notably, ChronoEdit-14B delivers clear improvements in Action Fidelity (4.01 *vs.* 3.76 for Qwen-Image and 2.88 for FLUX.1 Kontext [Dev]), while also maintaining competitive results in identity preservation (4.65) and visual & anatomical coherence (4.63). Among the three evaluation dimensions, action fidelity is particularly important as it directly

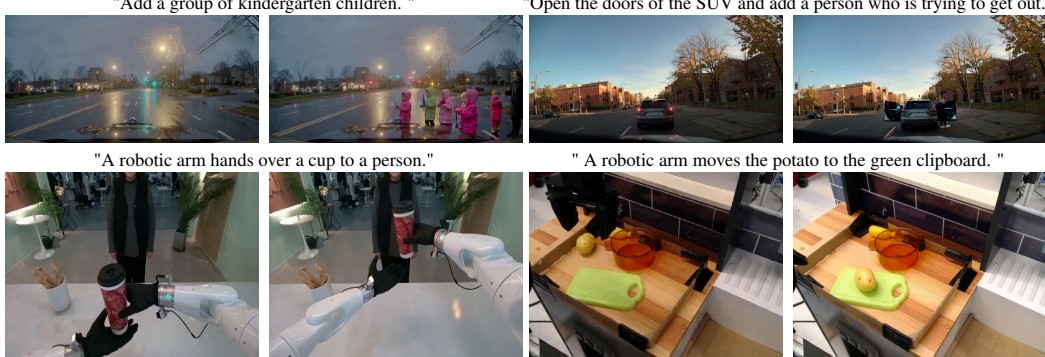

Figure 5: **Qualitative results on Physical-AI world simulation related tasks.** All results are generated by ChronoEdit-14B-Think. Each group shows a reference image (left) and the corresponding output (right). ChronoEdit produces edits that accurately follow the given instructions while preserving scene structure and fine details in Physical AI–related scenes.

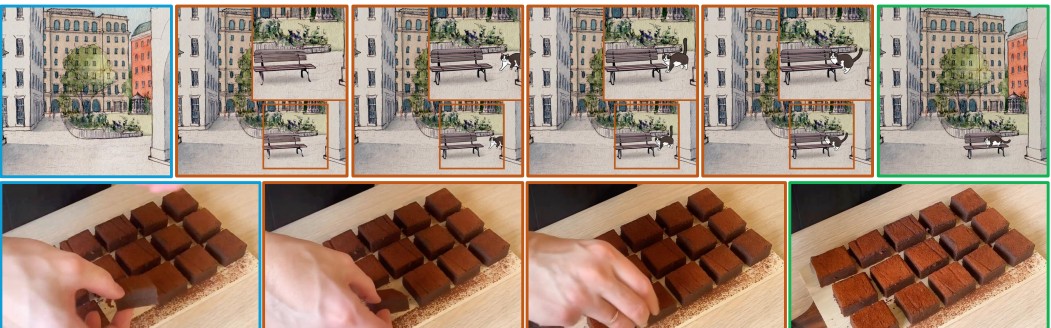

Figure 6: **Temporal reasoning trajectory visualization.** By retaining intermediate reasoning tokens throughout the entire denoising process, ChronoEdit-14B-Think is able to visualize its internal "thinking process" when performing edits. Sequences are shown from left to right: the reference image (blue box), decoded intermediate reasoning frames (orange boxes), and the final target frame (green box). Top example prompt: "Add a cat on the bench". Bottom example prompt: "Place a cake on a plate by hand".

reflects a model's ability to maintain physical consistency when performing edits involving real-world interactions. Even without Temporal Reasoning, ChronoEdit-14B benefits from its pretrained video prior, enabling it to achieve stronger results than all baseline image-editing models.

With Temporal Reasoning, ChronoEdit-14B-Think ($N_r = 10$) achieves a new state-of-the-art overall score of 4.53, with a particularly strong gain in Action Fidelity (4.31). This highlights the value of explicit Temporal Reasoning for edits that demand a deeper understanding of physical consistency. Notably, ChronoEdit-2B-think ($N_r = 10$) matches the performance of ChronoEdit-14B, falling only slightly short of ChronoEdit-14B-Think.

## 4.2 QUALITATIVE EVALUATION

**Comparison with Baselines.** We compare our approach against state-of-the-art image editing methods across a variety of challenging scenarios. As illustrated in Fig. 4, ChronoEdit consistently produces high-quality results, demonstrating competitive overall performance and, in particular, a clear advantage in action-oriented edits where precise modeling of dynamic poses and interactions is required. These results highlight the effectiveness of our video reasoning in handling complex, temporally grounded edits that are often difficult for conventional editing approaches.

**ChronoEdit on Physical AI Tasks** Figure 5 showcases ChronoEdit 's capability to address a broad spectrum of Physical-AI world simulation tasks. These results demonstrate the model's strong generalization across diverse domains of world simulation tasks, ranging from self-driving dynamics to robotic object manipulation.

**Temporal Reasoning Trajectory Visualization.** If the video reasoning tokens are fully denoised into a clean video, the model can illustrate how it "thinks" by visualizing intermediate frames as a

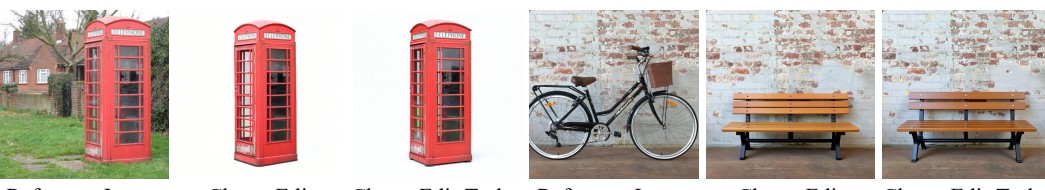

Reference Image    ChronoEdit    ChronoEdit-Turbo    Reference Image    ChronoEdit    ChronoEdit-Turbo

Figure 7: **Qualitative result of ChronoEdit-Turbo.** The lightweight ChronoEdit-Turbo (runtime 5.0s) achieves editing quality similar to ChronoEdit (runtime 35.3s) while offering improved efficiency. (Left: "Extract the red telephone booth in the image". Right: "Replace the bicycle in the image with a wooden park bench".)

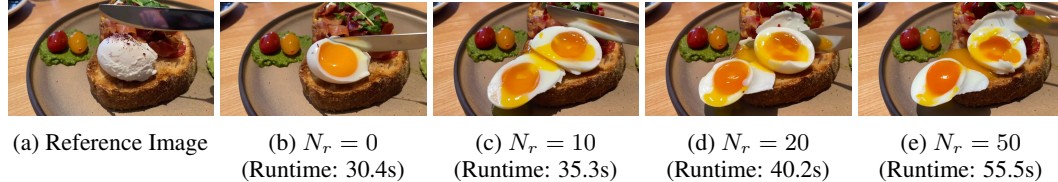

(a) Reference Image    (b) $N_r = 0$    (c) $N_r = 10$    (d) $N_r = 20$    (e) $N_r = 50$
(Runtime: 30.4s)    (Runtime: 35.3s)    (Runtime: 40.2s)    (Runtime: 55.5s)

Figure 8: **Qualitative ablation on video reason step $N_r$.** Empirically, we found that setting the reasoning timestep to $N_r = 10$ within a total of $N = 50$ sampling steps achieves performance that is comparable to using reasoning across the full trajectory. Example Prompt: "Halve the poached egg to reveal the yolk". Reported runtime is measured on Nvidia-H100 GPUs.

reasoning trajectory—though at the expense of slower inference. We present such a visualization in Fig. 6. As illustrated in the top row, when prompted to "add a cat on the bench", the model first synthesizes the bench and then anticipates the cat emerging from the corner and leaping onto it, composing the scene through a sequence of plausible intermediate states. Notably, an emergent capability of our approach is its ability to generate reasoning trajectory videos to realize edits. Even without exposure to training data where, for instance, a bench suddenly appears, the video model can still imagine and execute a plausible trajectory to accomplish the edit. In another example, the model correctly infers the stepwise process of placing a cake on a plate by hand. This deliberative trajectory reveals how the model perceives and interacts with the world in a coherent, physically grounded manner (see video visualization in Project Page).

**ChronoEdit-Turbo.** We further visualize the qualitative comparison of ChronoEdit and ChronoEdit-14B-Turbo in Fig. 7. Both ChronoEdit and ChronoEdit-Turbo successfully execute the edits with comparable visual fidelity, preserving scene structure and fine details. This demonstrates that the lightweight ChronoEdit-Turbo variant achieves editing quality comparable to that of ChronoEdit, while offering improved efficiency (5.0s *vs.* 30.4s runtime).

### 4.3 ABLATION STUDY

**Reasoning Timestep.** As discussed in Sec. 3.2, our model performs reasoning by traversing a sequence of intermediate states, thereby constructing a plausible temporal trajectory instead of directly regenerating the target image in a single step. Empirically, we found that setting the reasoning timestep to $N_r = 10$ within a total of $N = 50$ sampling steps achieves performance that is comparable to using reasoning across the full trajectory (Tab. 2), while reducing the overall computational overhead from 55.5s ($N_r = 50$) to 35.3s ($N_r = 10$), which is a small 4.9s increase compared to not using temporal reasoning (30.4s). An illustrative example is provided in Fig. 8, highlighting that shorter reasoning horizons are often sufficient to maintain fidelity while offering substantial efficiency gains.

Ablation studies on the benefits of video pretrained weights and encoding editing pairs design can be found in Appendix C.

### 5 CONCLUSION

We introduced ChronoEdit, a foundation model for image editing designed to enforce physical consistency. By repurposing a pretrained video diffusion model and introducing a temporal reasoning inference stage, our approach preserves coherence between input and edited outputs while producing plausible transformations. Extensive experiments demonstrate that ChronoEdit achieves state-of-the-art performance among open-source models.

## 6 ACKNOWLEDGEMENT

The authors would like to thank Product Managers Aditya Mahajan and Matt Cragun for their valuable guidance and support. We further acknowledge the Cosmos Team at NVIDIA, especially Qinsheng Zhang and Hanzi Mao, for their consulting of Cosmos-Pred2.5-2B. We also thank Yuyang Zhao, Junsong Chen, and Jincheng Yu for their insightful discussions. Finally, we are grateful to Ben Cashman, Yuting Yang, and Amanda Moran for their infrastructure support, especially in the period leading up to the deadline of this work.

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
