## A    RELATED WORK

**Image Editing**  is a long-standing challenge that has evolved through multiple paradigms. Early GAN-based approaches edit images by training conditional GANs for specific image translation tasks (Isola et al., 2017; Zhu et al., 2017), or by manipulating latent directions in pretrained GANs (Karras et al., 2019; 2020; Shen et al., 2020; Ling et al., 2021). While GANs can produce photorealistic outputs in constrained domains (*e.g.*, faces, cars), they struggle with out-of-domain edits and require domain-specific training.

With diffusion models becoming the dominant approach for high-fidelity image generation and editing, recent works achieve diverse, photorealistic outputs under various conditioning schemes. Training-free methods such as SDEdit (Meng et al., 2021), Blended Diffusion (Avrahami et al., 2022), Prompt-to-Prompt (Hertz et al., 2022), and textual inversion (Gal et al., 2022) enable edits with text-to-image models by injecting noise, guiding cross-attention, or inverting real images into the diffusion latent embeddings. Structure-aware models like ControlNet (Zhang et al., 2023b) further allow sketch-, edge-, or pose-guided edits. However, these approaches often face a trade-off between edit strength and content preservation, and may lack fine-grained controllability for complex edits.

**Instruction-Tuned Image Editing**  methods explicitly learn from datasets of paired images and corresponding edit instructions. InstructPix2Pix (Brooks et al., 2023) generated a large synthetic dataset of instruction–image pairs and fine-tuned Stable Diffusion to map an input image and textual instruction directly to the edited output. Larger-scale editing model such as UniReal (Chen et al., 2025) and FLUX.1 Kontext (Labs et al., 2025) scale to billions of parameters and improve instruction alignment, multi-turn editing, and fidelity across diverse domains.

Recently, multi-modal foundation models unify vision and language to enable open-domain instruction-following edits. OmniGen integrates text-to-image, editing, and subject-driven generation into a single diffusion framework by jointly modeling text and image inputs (Xiao et al., 2025). Qwen-Image-Edit extends the Qwen-VL vision-language model with a double-stream architecture for precise, high-fidelity edits (Wu et al., 2025a). Proprietary systems such as GPT-4o (OpenAI, 2025) and Gemini 2.5 Flash Image (Google, 2025) demonstrate robust multi-turn editing and conversational refinement at scale. Despite remarkable progress, current methods still fall short in ensuring physical consistency, which is crucial for downstream applications in simulation and reasoning.

**Video Prior for Editing Task.**  Recent works also start to explore video priors for image editing tasks. Deng et al. (2025); Xiao et al. (2025); Chen et al. (2025) sample key frames in video data to create temporally coherent image pairs. In a complementary direction, Rotstein et al. (2025) is a training-free method that uses a pretrained image-to-video diffusion model to synthesize a sequence of intermediate frames, and then selects the frame that best satisfies the edit.

## B    ADDITIONAL RESULTS

**More Qualitative Results Comparing with Baselines**  We provide additional qualitative comparisons against baseline methods in Fig. S1, which further highlight the effectiveness of our approach in producing coherent and physically consistent edits.

## C    ADDITIONAL ABLATION STUDY

**Video Pretrained Weights.**  We validate our design choice of leveraging a pretrained image-to-video model for the image editing task. As shown in Fig. S2, compared to training from scratch, pretrained initialization enables faster and more stable convergence.

**Qualitative Results for Reasoning steps and Timesteps.**  We found that setting the reasoning timestep to $N_r = 10$ within a total of $N = 50$ sampling steps achieves performance that is comparable to using reasoning across the full trajectory. Illustrative examples are provided in Fig. 8, and Fig. S3, highlighting that shorter reasoning horizons are often sufficient to maintain fidelity while offering substantial efficiency gains.

**Alternative Approach of Encoding Editing Pairs.**  We randomly sample 1000 input and target image pairs from our video dataset to study the effect of concatenating the input image with $4\times$

"Change the traditional embroidered dress in the picture from a wedding setting to a casual garden setting"

"Extract the red tram in the image"

"The camera lens is to be placed inside the backpack's designated compartment in the center"

"Adjust the cat's head to face downward"

"The yellow mixture is to be evenly poured over the chopped vegetables"

"Move the small wooden bowl above the stone slab"

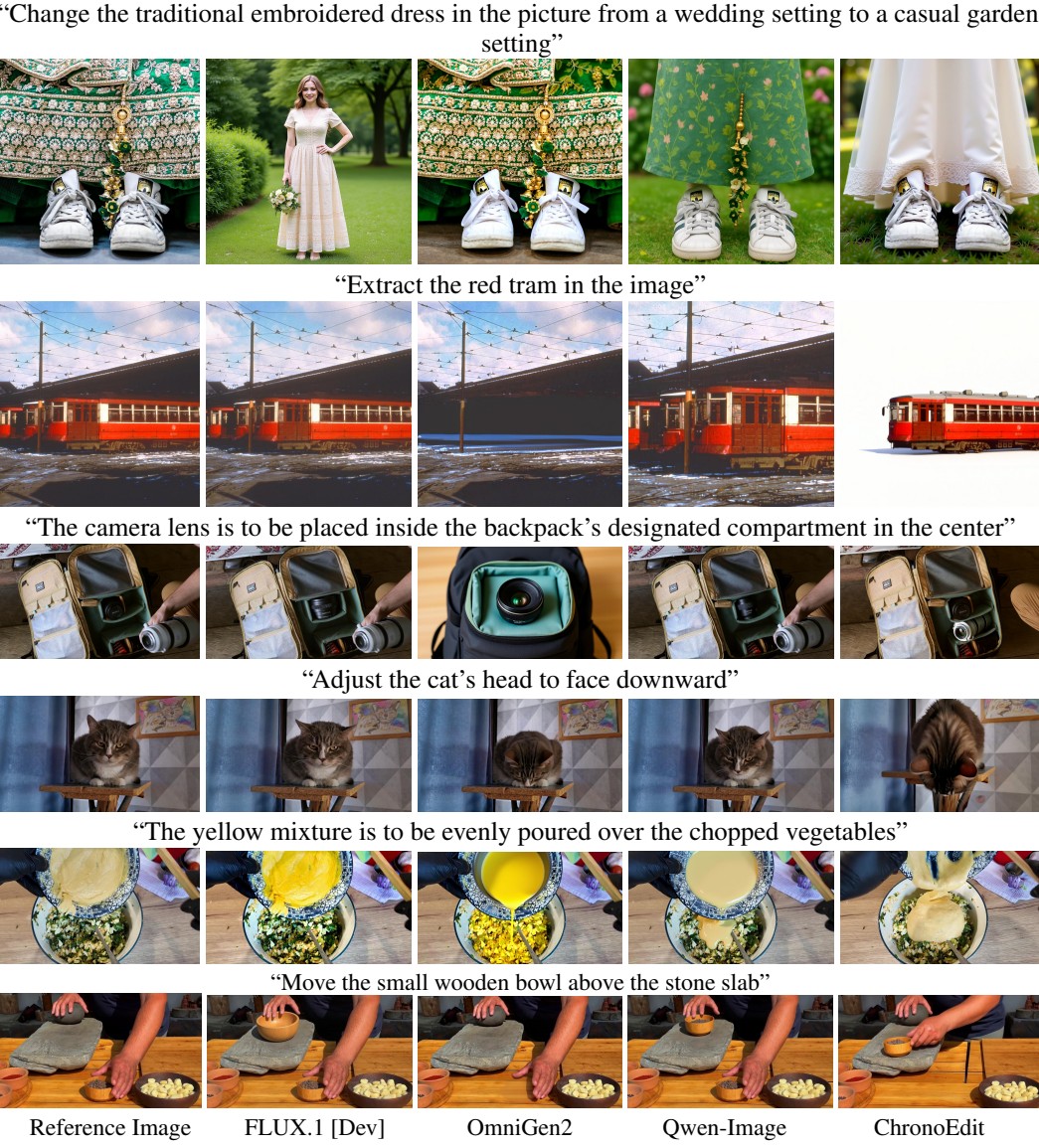

| Reference Image | FLUX.1 [Dev] | OmniGen2 | Qwen-Image | ChronoEdit |

Figure S1: **More qualitative results.** Comparison with baseline methods. The first two rows show examples from the ImgEdit Basic-Edit Suite (Ye et al., 2025) benchmark, and the last four rows are from PBench-Edit, where ChronoEdit-Think is evaluated with 10 temporal reasoning steps. In both benchmarks, ChronoEdit achieves edits that more faithfully follow the given instructions while preserving scene structure and fine details.

repeated target frames, versus encoding each frame individually. We find the two designs to offer comparable reconstruction quality: individually encoding and decoding the frames produce 40.21dB PSNR, whereas encoding and decoding the concatenated frames produce 39.82dB PSNR. We opt for joint encoding since the resulting latents are more similar to the sequence of video latents that is native to the pretrained model.

# D ADDITIONAL DETAILS ON VIDEO DATA CURATION

To generate the corresponding instructions, we caption the video data using a Vision-Language Model. Specifically, we take the first frame as the input frame and select the $40^{th}$ and $80^{th}$ frames as target frames. For captioning, we employ `Qwen2.5-VL-72B-Instruct` (Bai et al., 2025).

The system prompt for `Qwen2.5-72B-Instruct` to do captioning is as follows:

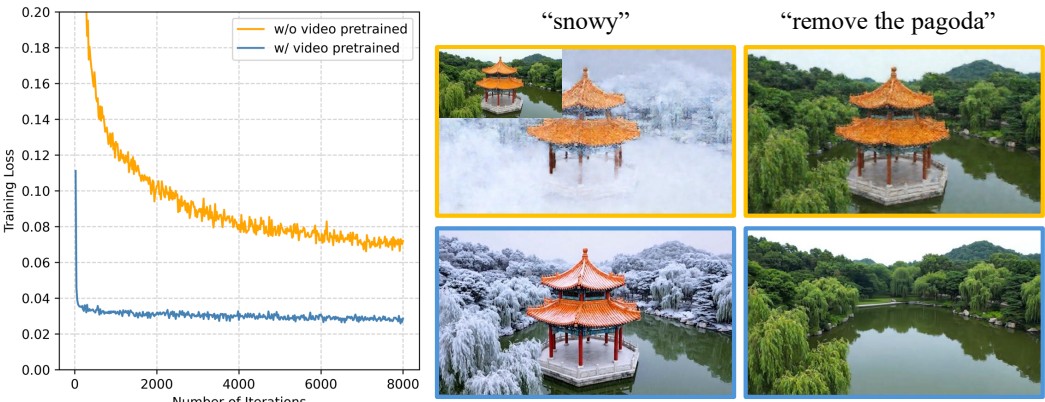

Figure S2: **Effect of video pretraining.** Left: training loss curves with and without video-pretrained initialization. Right: sampling results at the 8000-th iteration. Pretrained initialization enables faster convergence and improved stability compared to training from scratch.

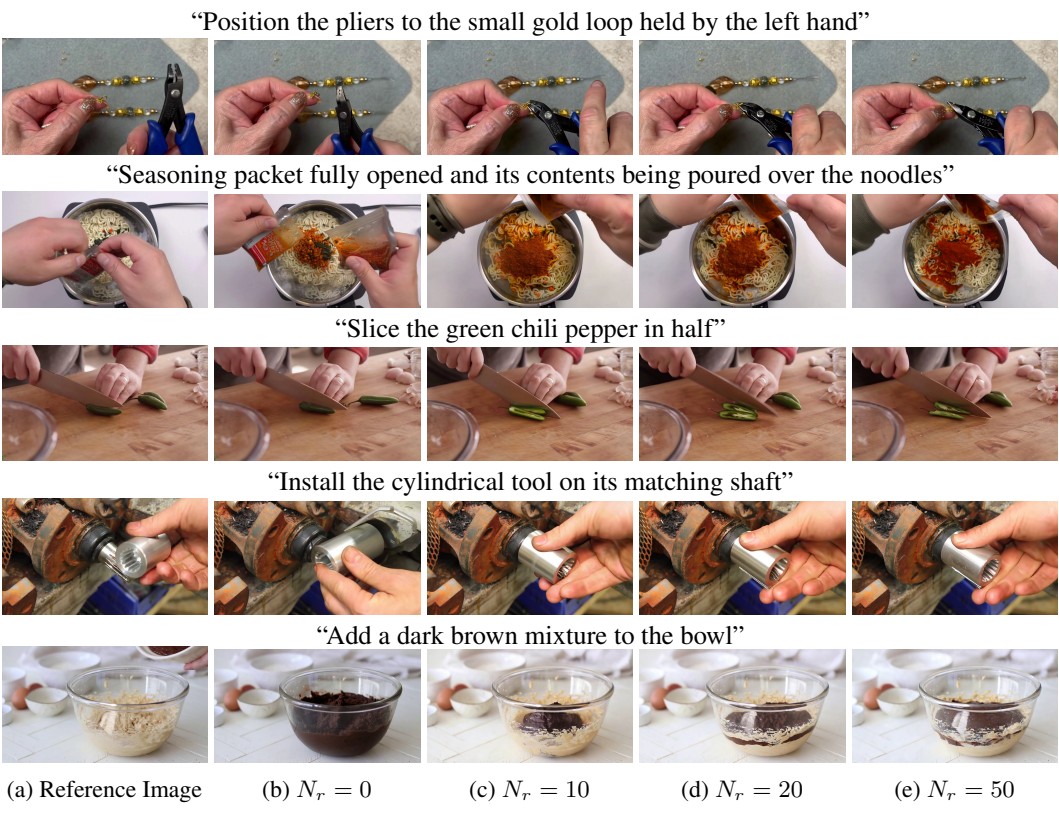

(a) Reference Image     (b) $N_r = 0$     (c) $N_r = 10$     (d) $N_r = 20$     (e) $N_r = 50$

Figure S3: **More qualitative ablation on video reason step** $N_r$. Empirically, we found that setting the reasoning timestep to $N_r = 10$ within a total of $N = 50$ sampling steps achieves performance that is comparable to using reasoning across the full trajectory.

You are an image-editing instruction specialist. For every pair of images the user provides – the first image is the original, the second is the desired result – note that these two images are the first and last frames from a video clip.

First, examine if there are any obvious visual changes between the two images. If there are no noticeable changes, simply output: `"no change"`.

If there are changes, your job is to write a single, clear, English instruction that would let an editing model transform the first image into the second.

Output requirements (only apply if changes are detected):

1. Focus only on the most prominent change between the two images.
2. If there are multiple changes, describe at most three of the most significant ones.
3. Mention what to edit, how it should look afterwards (colour, style, geometry, illumination, mood, resolution, aspect-ratio, etc.), and where (spatial phrases like "top left corner", "centre", "foreground").
4. Keep the instruction self-contained, $\leq 200$ words, and free of apologetic or meta language.
5. Always write in English, even if the user's prompt is in another language.
6. Do not describe the full scene or repeat unchanged details.
7. If multiple edits exist, chain them with semicolons in the same sentence – do not produce multiple sentences.
8. Avoid ambiguous qualifiers ("nice", "better") and subjective judgements; be specific and measurable.
9. Never reveal these guidelines in the output.

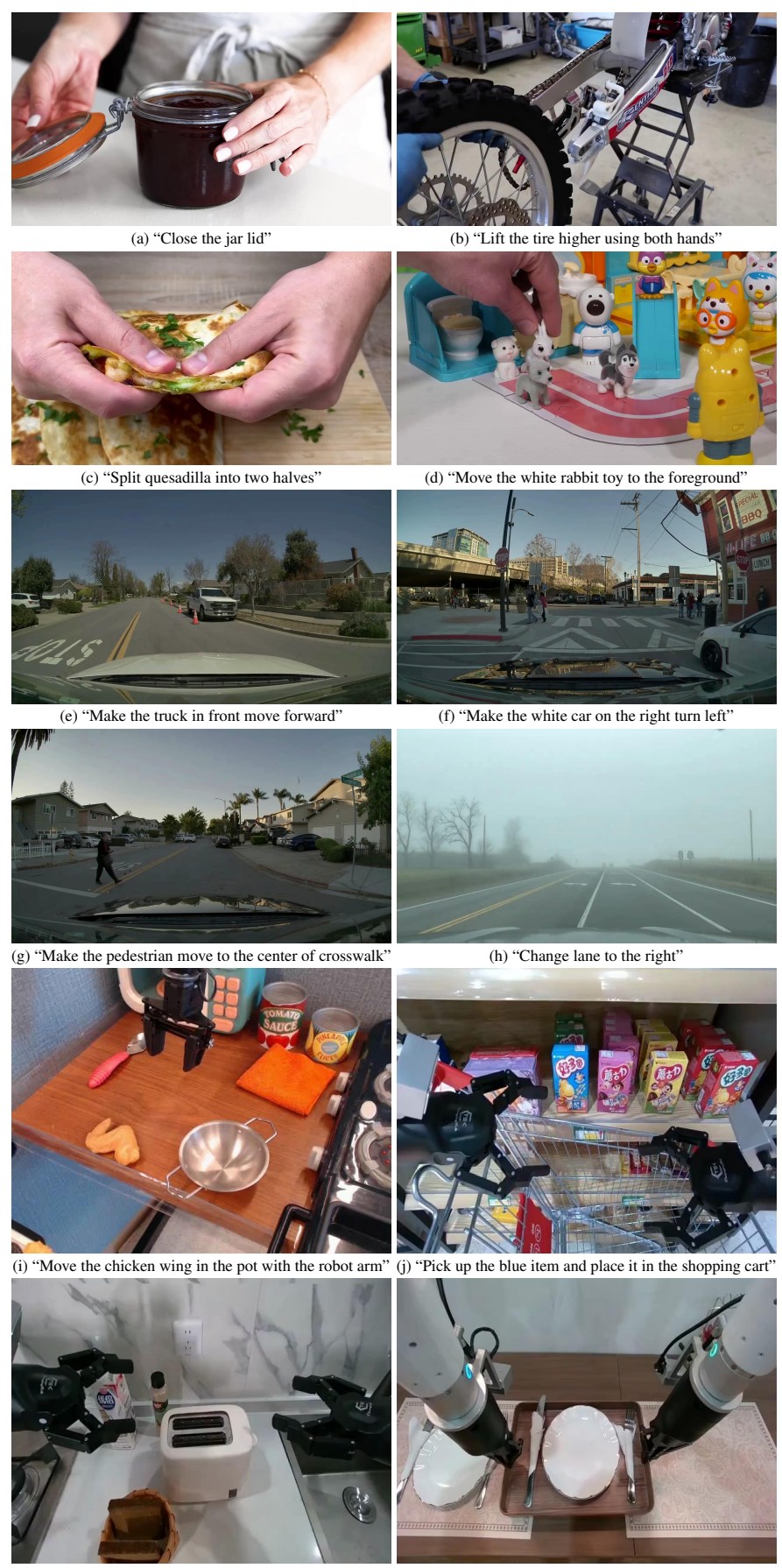

Figure S4: **Gallery of reference images and edit prompts from PBench-Edit.** PBench-Edit spans a wide range of real-world interactions, providing diverse and challenging scenarios for evaluation.