# OpenReview forum: "ChronoEdit: Towards Temporal Reasoning for In-Context Image Editing and World Simulation"
_ICLR.cc/2026/Conference — ICLR 2026 Poster_

### Official Review · Reviewer_kjZH · 2025-10-20

**Soundness:** 4
**Presentation:** 3
**Contribution:** 3
**Rating:** 8
**Confidence:** 5

**Summary:**

This paper proposes modeling the physical realism of image editing using video generation models. By leveraging the strong temporal, physical, and motion consistency capabilities of video generation models, the approach achieves impressive editing results. Furthermore, the introduction of a temporal reasoning token to simulate intermediate video frames is a very intuitive idea, and the experimental results are remarkable.

**Strengths:**

1. Utilizing video generation model to model the image editing task, which introducing great physical prior, achieving great results.
2. The Temporal Reasoning Token simulate the intermidiate step of video and it fills the gap of modeling intermediate changes in image editing, resulting in stronger interpretability.
3. After distilling, the result is still great and the speed is nearly comparable with image editing model.

**Weaknesses:**

I do not have many concerns regarding the content of the paper itself. However, I am curious about how the video generation based model would perform in multi-turn editing scenarios involving different user instructions, where each round would introduce an additional input frame, which is an interesting attempt for future development.
Additionally, I want to know the true performance of trained video generative model in video generation. This would provide insights into how demanding the requirements are for training video generation models to achieve high-quality image editing. Please show the result in video generation benchmark like VBench series.

**Questions:**

See the weakness.

---

> ### Author Response · Authors · 2025-11-19
>
> **[Q1: multi-turn editing]**
>
> We thank the reviewer for the insightful feedback. As shown in Fig. S2, ChronoEdit naturally supports multi-turn editing while maintaining strong visual consistency across successive modifications. When users apply a sequence of edits, the model preserves object identity, style, and structure, enabling coherent transformations without drift or quality degradation. This demonstrates ChronoEdit’s ability to handle complex, iterative editing workflows. We agree with the reviewer that explicitly introducing an additional input frame at each round is an interesting avenue for future research, which we plan to explore.
>
> **[Q2: video results]**
>
> We provide qualitative visualizations of the video reasoning trajectory on our supplementary website. The generated videos are visually comparable to the base video diffusion model, preserving its prior knowledge. Our focus is on high-quality image editing rather than conventional video generation. To optimize for editing efficiency, we employ specialized designs, such as shortening the temporal latent to 8 tokens. Because of these modifications, a direct quantitative comparison with standard bi-directional video generation models would be unfair. Therefore, we show qualitative visualizations but do not evaluate quantitatively against base video diffusion models, and we primarily benchmark performance on editing tasks.

---

### Official Review · Reviewer_rm73 · 2025-10-23

**Soundness:** 2
**Presentation:** 3
**Contribution:** 2
**Rating:** 2
**Confidence:** 3

**Summary:**

The authors propose a method termed ChronoEdit using latent diffusion/flow video models for editing images based on text prompts and a reference image. Three variants of the method are proposed in the paper. (1) ChronoEdit - generations of the edited image as a video having two frames (the reference image and the output image); (2) ChronoEdit-Think - where additional "reasoning" frames/tokens are added between these two frames in order to model better physical consistency across time; (3) ChronoEdit-Turbo - distilled version of ChronoEdit for few-step sampling. The paper also presents PBench-Edit, a new benchmark for image editing designed to assess editing in physically grounded contexts. The authors compare the 3 variants to various baselines from the literature on two benchmarks and show an ablation study for various aspects of their method.

**Strengths:**

* The main novelty of this paper, as I understand it, since I am not familiar with the literature in this area, resides in the idea to frame the problem of image editing based on text prompts as a video generation task using diffusion/flow models. By doing so, one can get better physical consistency. Indeed, I find this idea novel and interesting.
* To gain better control and interpretability, another novel part of the method is to introduce the so-called reasoning tokens, which show that the authors took an extra step in the modeling process of the problem.
* The authors introduce a new benchmark, which I suppose can be valuable for research in this area.
* The method seems to improve over baseline methods on both benchmarks, both quantitatively and qualitatively.
* For the most part, the paper is written clearly.

**Weaknesses:**

* While I do appreciate the idea of using video generation models for image editing and the other improvements to the method, in terms of the overall contribution and novelty, to me, this work seems limited. To properly assess its contribution, I believe it will be valuable if the authors could further elaborate on how this work lays a new foundation for future models in this area and if it opens new avenues of research.
* Regarding the experiments:
  - To me, the quantitative results are not clear enough. Specifically, are the numbers in the table bounded between [0,5]? If not, are they bound in another region? How exactly are they measured? In addition, in order to understand the significance of the results, std information should be reported.
  -  The authors present only successful cases for their model, but what are some failure cases of it? Are there common situations in which it systematically fails?
  - One experiment I found missing is showing the performance of the method when using out-of-the-box video generation models (without training), where the first and last frames are set as in this paper.
  - Regarding reproducibility, code wasn't provided. For me, this is a weakness in an empirical paper.
* Minor:
  - In Fig. 3, what is the prompt?
  - In line 311, do you mean $N=8$?
  - It is not clear which version of ChronoEdit, ChronoEdit-Turbo distills.

For now, my tendency is to reject the paper; however, as I am not an expert in this field, I would like to see the author's response, assessment of my fellow reviewers, and the AC before making a final decision.

**Questions:**

* In line 201, how is it ensured that F', c, and w are integers?
* Was the model from which you generated the training dataset ("video data curation") trained (either in the pre-training stage or fine-tuning) on PBanch?
* All baseline methods seem recent; why is there such a gap in the number of parameters? Specifically, I assume that at least part of the baseline methods also use latent diffusion/flow models. What causes the difference then? And can you evaluate your method using exactly the same network and number of parameters?

---

> ### Author Response · Authors · 2025-11-19
>
> **[Q1: contribution to future models]**
>
> ChronoEdit introduces a novel paradigm by reframing image editing as a video generation problem, where the temporal structure acts as a prior to enforce physical consistency—capabilitiy absent in existing image editing methods (see Fig. 2).
>
> Our framework provides a general recipe for converting any pre-trained video diffusion model into a powerful image editor. As video models advance, this design becomes a reusable foundation: improvements in video modeling directly translate into stronger editing capabilities without requiring architectural redesign.
>
> Moreover, our temporal reasoning mechanism enables “chain-of-frame” thinking, offering interpretability and a new way for models to reason about gradual transformations, object interactions, and scene dynamics. This opens research directions beyond prior image editing approaches. We believe ChronoEdit delivers not only a practical state-of-the-art system but also a conceptual foundation for future work in physically consistent and temporally coherent image editing.
>
> **[Q2: quantitative results]**
>
> We follow the standard evaluation protocol adopted by recent image-editing benchmarks. The scores in our tables are indeed bounded within [1, 5], where 1 denotes the worst and 5 the best performance. These scores are generated using GPT-4.1, which evaluates instruction adherence, edit quality, and detail preservation based on the input–output image pair and edit instruction. We strictly adhere to the official evaluation procedure provided in the benchmark (https://github.com/PKU-YuanGroup/ImgEdit). The full evaluation code is publicly available for full transparency and reproducibility.
>
> Regarding standard deviation: the GPT-based assessment produces deterministic scores under fixed system prompts. Consequently, existing benchmarks and prior works do not report standard deviations. For consistency and fair comparison, we follow the same reporting convention.
>
> **[Q3: failure cases]**
>
> We appreciate the reviewer’s interest in understanding the limitations of our method. While ChronoEdit performs strongly across diverse tasks, it still faces limitations in certain challenging scenarios that require more advanced reasoning. In particular, we found that the model struggles with tasks involving complex spatial understanding and multi-step logical reasoning (see Fig. S4 in the revised manuscript). This limitation is not unique to ChronoEdit—most existing state-of-the-art vision-editing models, including GPT Image 1 and Qwen-Image, face similar challenges.
>
> We believe the primary cause is the lack of targeted, high-quality training data that explicitly captures these reasoning-intensive cases. Addressing this gap and improving the model’s reasoning capabilities remain important directions for future work.
>
> **[Q4: experiment on out-of-the-box video generation models]**
>
> Thank you for pointing this out. We have now conducted the experiment by running the pretrained video generation model out of the box, simply using the last frame of image-to-video model as an edited image. We now include a side-by-side comparison in Fig. S3 of revised paper. The raw video model fails on most editing tasks—including Replace, Add, and Style Transfer. In contrast, our image–video cotraining design enables the same pretrained model to operate as a coherent and precise image editor. This demonstrates the importance of proper training recipes and architecture design for transforming a video model into a physically grounded image editing system.
>
> **[Q5: reproducibility]**
>
> We agree that reproducibility is essential. We are fully committed to open-sourcing our work and will release the complete training and inference code, evaluation scripts, and model checkpoints upon publication. This will allow the community to reproduce all results and build upon our method.
>
> **[Q6: prompt in Fig. 3]**
>
> The prompt in Fig. 3 is: “Add a coffee mug on the table in the foreground.” We have added this prompt in the revised paper.
>
> **[Q7: line 311]**
>
> We clarify that T=8 represents the total number of tokens used for video reasoning. This includes the input image token, six video tokens, and one edit image token. We have revised this typo in the paper.
>
> **[Q8: ChronoEdit-Turbo]**
>
> ChronoEdit-Turbo is distilled directly from the full ChronoEdit model. Importantly, it also supports temporal reasoning (Think mode), preserving the core capabilities of the original system while offering significantly improved efficiency. We will clarify this in the final paper.

---

> > ### Author Response · Authors · 2025-11-19
> >
> > **[Q9: line 201, how is it ensured that F', c, and w are integers]**
> >
> > The values $F^\prime$, $C$, $h$, and $w$ are guaranteed to be integers because they are determined by the architecture of the pretrained Wan2.1 VAE, not chosen arbitrarily. The VAE uses fixed downsampling factors in both the spatial and temporal dimensions. Specifically, Wan2.1 applies:
> > - Temporal downsampling by a factor of 4, with padding/striding chosen such that $(F - 1)$ is always divisible by 4, ensuring $F^\prime = \frac{F - 1}{4} + 1$ is an integer for all supported input lengths.
> > - Spatial downsampling by a factor of 8 in both height and width, and the model accepts only resolutions where $H$ and $W$ are multiples of 8, ensuring $h = H / 8$, $w = W / 8$ are integers.
> > - The channel dimension $C = 16$ is fixed by the pretrained VAE architecture.
> > In short, all valid inputs to the Wan2.1 VAE satisfy the divisibility requirements imposed by its architecture, so the resulting latent dimensions are always integers by construction.
> >
> > **[Q10: training dataset]**
> >
> > No, the model we used for generating the training dataset in the video data curation stage was not trained or fine-tuned on PBench-Edit. PBench-Edit is an entirely independent evaluation benchmark specifically designed to assess editing performance in physically grounded scenarios. Our training data and models do not include any images or annotations from PBench, ensuring a clean separation between training and evaluation.
> >
> > **[Q11: the parameter gap among baseline methods]**
> >
> > The parameter gaps primarily reflect the rapid evolution of pretrained generative backbones used in prior work. Earlier editing methods such as MagicBrush and Instruct-Pix2Pix are all built on Stable Diffusion, which contains ~0.9B parameters. More recent approaches rely on significantly larger and more capable backbones: FLUX-based editors use a 12B DiT-style model, and the newest Qwen-Image-Edit adopts a 20B backbone. These differences naturally lead to a wide variation in parameter counts across baselines.
> >
> > In contrast, our method is built on Wan2.1-I2V-14B, a pretrained video diffusion model with a DiT architecture comparable in size to FLUX but fundamentally different in design. Leveraging a video backbone—and showing that it can be converted into a powerful image editor—is one of our key contributions.
> >
> > There is currently no pretrained Wan model available at 12B or 20B scale, so it is not possible to evaluate ChronoEdit using exactly the same model sizes as FLUX or Qwen-Image-Edit. Nevertheless, our results show that ChronoEdit (14B) outperforms both FLUX-based editors (12B) and Qwen-Image-Edit (20B), demonstrating that our performance gains stem from the proposed framework rather than parameter count alone.

---

> > > ### Comment · Reviewer_rm73 · 2025-11-24
> > >
> > > I thank the authors for the comments and clarifications. I believe that the additional experiments, including out-of-the-box generation and failure cases are valuable additions to the paper. Hence, following the rebuttal, I will raise my score to 6.

---

### Official Review · Reviewer_BTgq · 2025-11-01

**Soundness:** 3
**Presentation:** 3
**Contribution:** 3
**Rating:** 8
**Confidence:** 3

**Summary:**

The paper introduces ChronoEdit, a novel framework designed to instill physical consistency in generative image editing, a capability deemed crucial for "world simulation" applications (e.g., robotics, autonomous driving). ChronoEdit achieves this by reframing image editing as a two-frame video generation problem, thereby leveraging the learned temporal prior from pre-trained video generative models. For enhanced coherence, the framework features a Temporal Reasoning Stage (ChronoEdit-Think) during inference. Here, "reasoning tokens" (imagined intermediate video frames) are jointly processed with the input to implicitly constrain the final edit trajectory to one that is physically plausible. The authors validate their approach with the new PBench-Edit benchmark, demonstrating SoTA performance in both visual quality and physical realism over existing methods.

**Strengths:**

1. The idea of repurposing the powerful temporal mechanisms of video generation models to solve a fundamental deficiency (physical inconsistency) in static image editing is a good idea, providing a principled method for incorporating dynamic laws.
2. By focusing on physical consistency, ChronoEdit addresses an important bottleneck for real-world applications like autonomous systems, where geometric or physical inconsistencies are unacceptable.
3. The Temporal Reasoning Stage is cleverly implemented to optimize efficiency by limiting the reasoning steps ($N_r$) and discarding the tokens early in the denoising process. Furthermore, the visualized reasoning trajectory offers a degree of interpretability into the model's "thinking process."
4. The creation of the PBench-Edit benchmark is a valuable contribution, establishing a much-needed standard to evaluate models on physical and temporal coherence, pushing research beyond purely aesthetic metrics.
5. Figure 4 looks great

**Weaknesses:**

1. Despite efforts to optimize, the ChronoEdit-Think variant still introduces a measurable inference overhead compared to non-reasoning baselines.
2. The framework’s success depends entirely on the video model's ability to perfectly encode and enforce physical laws within its high-dimensional latent space. If the latent representation merely captures statistical correlations of motion rather than strict physical principles, errors in complex or novel dynamics will inevitably persist.

**Questions:**

1. What editing operations fundamentally cannot benefit from this framework?
2. Is rejection sampling effective for generating physically plausible results?

---

> ### Author Response · Authors · 2025-11-19
>
> **[Q1: ChronoEdit-Think inference overhead]**
>
> We agree that ChronoEdit-Think introduces additional computation. However, full-step temporal reasoning is not required. As shown in Fig. 7, the video-reasoning tokens can be selectively applied, and dropping them for most sampling steps significantly reduces overhead with minimal performance impact. Using only 10 reasoning steps out of 50 already matches the quality of full temporal reasoning while adding just ~5 s of runtime (ChronoEdit: 30.4 s → ChronoEdit-Think with $N_r=10$: 35.3 s). In addition, the distillation approach described in Sec. 3.4 can further improve efficiency by a large margin. Thus, while some overhead is expected, our design keeps it modest and controllable without sacrificing performance.
>
> **[Q2: reliance on the video model]**
>
> The framework's reliance on the video model for encoding physical laws is a valid concern. We acknowledge that no current generative model perfectly encapsulates the full spectrum of physical principles. However, ChronoEdit's advantage is two-fold:
>
> - Implicit Physical Reasoning: By formulating image editing as a constrained video generation problem, the model is inherently forced to traverse a physically plausible trajectory from the initial state (input image) to the final state (edited image). The video model, even if only capturing strong statistical correlations, is significantly better at generating temporally coherent and consistent transitions (e.g., motion, lighting, structural changes) than a single-step image-to-image translation model.
>
> - Enforced Temporal Consistency: The ChronoEdit loss function explicitly encourages the intermediate frames (the "thought process") to align with a consistent temporal flow. This temporal consistency acts as a regularization mechanism, steering the generation away from purely statistical, physically contradictory outcomes that might otherwise be produced by non-reasoning image models. The intermediate frames serve as a dynamic constraint, implicitly enforcing a form of "physical common sense" derived from the model's vast training data on real-world videos.
>
> While errors in complex, non-canonical dynamics might still occur, our method significantly mitigates them compared to standard image editing. The temporal trajectory provides a path of least resistance in the model's latent space, favoring physically smoother and more plausible transitions. Our quantitative results on physics-sensitive tasks (Tab. 2) validate that this approach yields a substantial improvement in physical coherence compared to baselines. Notably, the video generative model serves as our prior: as video models become better at enforcing physical laws, our method correspondingly benefits, enhancing its physical reasoning capabilities.
>
> **[Q3: editing operations cannot benefit from this framework]**
>
> Our framework is broadly effective across diverse editing scenarios, as evidenced by Tab. 1, where we achieve state-of-the-art results on 9 common editing tasks—even outperforming strong proprietary systems such as GPT-Image 1 and FLUX.1 Kontext [Pro]. In practice, we find very few editing operations that do not benefit from our method.
>
> Edits that require motion- or action-level changes benefit the most, since our temporal reasoning directly enforces physical consistency—capabilities that standard image editing pipelines lack.
>
> The only category that shows marginal additional gains is pure style transfer, where the desired change largely preserves geometry and does not rely on temporal coherence. Even in this case, our framework remains competitive.
>
> **[Q4: rejection sampling]**
>
> We agree with the reviewer that rejection sampling is a promising complementary technique to help encourage physical plausibility. While it is orthogonal to our core method, it could be integrated as a post-hoc filtering step to further improve physical coherence. We view this as a valuable extension, and plan to explore it in future work.

---

### Official Review · Reviewer_dEVi · 2025-11-02

**Soundness:** 3
**Presentation:** 2
**Contribution:** 2
**Rating:** 4
**Confidence:** 2

**Summary:**

This manuscript proposes a ChronoEdit method for image editing, which reframes the editing problem as a video generating problem. It introduces a temporal reasoning mechanism at inference to perform editing, wherein reasoning tokens are designed to imagine plausible editing trajectory. A novel benchmark dataset is proposed to evaluate the proposed method.

**Strengths:**

-	The idea of solving image editing using video generation models is interesting.
-	Experimental results on the benchmark datasets show good performance.

**Weaknesses:**

-	The proposed method is prone to the video generation models. Is the upper bound of the proposed method is limited to the performance of the video generation models?
-	The computational complexity of the proposed method vs. other image editing methods is expected to be justified.
-	The ambiguity of the “edited” image. As mentioned by the author in L216-232, the author formulates the image editing as a T-frame video generation problem, wherein 0- and T-th frames are defined as the input and edited image. Can the (T-1)-th frame or other frames also be considered as the “edited” image? If not, what are the key differences between the (T-1) and T frames?
-	There are methods [1-3] that apply chain-of-thought techniques to the image editing problem, which also attempt to add an intermediate process into the editing problem. Can you discuss and justify the proposed paradigm vs these methods?

[1]. Enhancing Image Editing with Chain-of-Thought Reasoning and Multimodal Large Language Models
[2]. ReFocus: Visual Editing as a Chain of Thought for Structured Image Understanding

**Questions:**

See the weakness

---

> ### Author Response · Authors · 2025-11-19
>
> **[Q1: upper bound of the proposed method]**
>
> We reframe the video generative model as an image editing model by introducing image–video co-training (Sec. 3.2). In our approach, the pre-trained video model serves as a prior rather than a strict performance upper bound. As shown in Fig. S3 of the revised paper, our method demonstrates editing capabilities beyond the base video model. For example, the pre-trained video model struggles with object replacement, texture modification, and style transfer. While it can add an object to some extent, it often fails to maintain content consistency (e.g., in row 4 of Fig. S3, the camera moves and the existing tree in the reference image disappears). In contrast, our method preserves consistency and enables precise edits, indicating that its performance is not inherently limited by the video model.
>
> **[Q2: computational complexity]**
>
> We benchmarked ChronoEdit’s computational complexity against other image editing methods on an NVIDIA H100 GPU, using 35 inference steps and a resolution of 720×1280 by default. Despite leveraging a pre-trained video generative model, our method remains efficient compared to existing open-source approaches, and even with temporal reasoning (ChronoEdit-Think), the computational cost is manageable.
>
> With the distillation technique described in Sec. 3.4, ChronoEdit-Turbo achieves 5.95 s per image—a ~4× speed-up over the original ChronoEdit (23.75 s) while maintaining comparable visual quality. It is also ~14× faster than the most competitive open-source method (Qwen-Image), demonstrating that our approach delivers both high efficiency and strong performance.
>
> | Model                       | Model Size | Runtime  | Peak GPU Memory |
> |----------------------------|------------|----------|-----------------|
> | OmniGen2                   | 7B         | 22.93s   | 18GB         |
> | Step1X-Edit                | 19B        | 16.35s   | 42GB         |
> | FLUX.1 Kontext [Dev]       | 12B        | 34.28s   | 36GB         |
> | Qwen-Image                 | 20B        | 84.37s   | 60GB         |
> | ChronoEdit                 | 14B        | 23.75s   | 34GB            |
> | ChronoEdit-Turbo (8 steps) | 14B        | 5.95s    | 34GB            |
> | ChronoEdit-Think       | 14B    | 38.18s | 38GB       |
> | ChronoEdit-Think-Turbo (8 steps) | 14B | 12.34s | 38GB |
>
> **[Q3: the edited image]**
>
> Our model is image co-trained: during image training iterations, we set the target frame at the last latent timestamp (practically, we set the number of latent tokens to 8), which aligns with the length of the video training data. Empirically, using the T-th frame produces the best results.
>
> The key distinction is that frame T represents the fully realized edit that best satisfies the user’s instruction, whereas earlier frames (e.g., T-1) correspond to intermediate states along the editing trajectory. For example, in Fig. 6, to place a cake on a plate by hand, frame T shows the final result with the cake correctly placed on the plate, while the intermediate frames depict the transition steps (hand lowering → cake on the plate → hand moving away). These intermediate frames ensure a smooth transition from the source image to the target edit and provide interpretability by showing how the model arrives at the final edit.
>
> **[Q4: compare to CoT-style works]**
>
> We appreciate the reviewer pointing out related CoT-style works. We will add the discussion in the next version. Here we provide some brief discussion about difference between proposed paradigm and both [1] and [2].
>
> [1] performs chain-of-thought reasoning in an LLM to guide an external diffusion model. The intermediate steps are linguistic/logical descriptions or localized masks produced by the LLM, not transformations occurring inside the generative model itself. In contrast, ChronoEdit introduces a Chain-of-Frame reasoning process inside the video diffusion model, enabling physically consistent, temporally grounded transformations. Our intermediate “steps” are latent video frames, not text-based reasoning, and they directly encode object motion, camera movement, and geometry transitions—capabilities beyond LLM-driven CoT.
>
> [2] uses multimodal LLMs to generate Python code that performs sequential operations (drawing boxes, cropping, masking) as part of a visual understanding pipeline. These “edits” serve only as auxiliary cues for recognition, not as high-fidelity image editing. Their goal is not to produce a photorealistic edited image, but to enhance reasoning. ChronoEdit’s target is entirely different: high-quality, physically consistent image synthesis.
>
> In summary, CoT techniques in [1–2] operate at the LLM or tool-calling level and do not modify the generative process of a video diffusion model. ChronoEdit introduces a new temporal reasoning paradigm inside the diffusion backbone, enabling continuous, physically consistent edit trajectories. Therefore, our method is fundamentally distinct in both mechanism and objective.

---

> > ### Comment · Reviewer_dEVi · 2025-11-19
> > **Address my concerns**
> >
> > Thanks to the authors' detailed response. All my concerns are addressed, and I'd like to raise my rating.

---

### Author Response · Authors · 2025-12-02
**Rebuttal Summary for AC**

## 1. Main strengths (across reviews)

- **New paradigm**
  ChronoEdit reframes image editing as a constrained video generation problem, using temporal priors from video models to enforce physical and geometric consistency in edits (`dEVi`, `rm73`, `BTgq`, `kjZH`).

- **Temporal reasoning and interpretability**
  The use of “reasoning tokens” (intermediate latent frames) provides a chain-of-frame trajectory that makes edits more physically plausible and interpretable (`dEVi`, `BTgq`, `rm73`, `kjZH`).

- **Empirical performance**
  ChronoEdit and its variants achieve strong quantitative and qualitative performance on two benchmarks, including state-of-the-art results on physics-sensitive editing tasks and on the new PBench-Edit benchmark (`BTgq`, `rm73`, `kjZH`).

- **Benchmark contribution**
  PBench-Edit is recognized as a valuable benchmark for physical realism and temporal coherence in editing, going beyond purely aesthetic metrics (`dEVi`, `BTgq`, `rm73`).

---

## 2. Additional experiments and clarifications during rebuttal

### 2.1 Comparison with pretrained video model (requested by Reviewer `rm73`, `dEVi`)

New experiment added: **Comparison with Pretrained Video Model** (Fig. S3)

- (`rm73`) We show that the raw video model—used directly without ChronoEdit—fails on Replace, Add, and Style tasks (Fig. S3).
- (`dEVi`) On the same backbone, ChronoEdit succeeds, demonstrating that our method extends the video model’s capabilities and is not upper-bounded by it.

### 2.2 Multi-turn editing (requested by Reviewer `kjZH`)

New experiment added: **Multi-turn Editing** (Fig. S2)

- We show in Fig. S2 that ChronoEdit supports sequential editing steps while preserving identity, geometry, and style across turns.
- This directly addresses concerns about model behavior when edits accumulate (`kjZH`).

### 2.3 Efficiency clarifications (requested by Reviewer `dEVi`, `BTgq`)

- (`dEVi`) We include a complete report comparing ChronoEdit to baselines in terms of size, runtime, and peak memory.
- (`BTgq`) We show that selective reasoning reduces the reasoning (“Think”) overhead to about 5s (for 10/50 steps), while retaining full quality. ChronoEdit-Turbo maintains reasoning capability with roughly a 4× speed-up.

### 2.4 Failure cases (requested by Reviewer `rm73`)

New experiment added: **Failure Cases** (Fig. S4)

- We added examples (Fig. S4) where ChronoEdit struggles, primarily on tasks requiring complex logical or spatial reasoning.

---

## 3. Individual reviewer updates

- **Reviewer `dEVi`**
  - Rating: 4 → 6
  - Response: All concerns were resolved in detail. The reviewer explicitly stated that all issues were addressed and updated the score to 6 on **19 Nov 2025**.

- **Reviewer `BTgq`**
  - Rating: 8 (unchanged)

- **Reviewer `rm73`**
  - Rating: 2 → 6
  - Response: The reviewer explicitly stated that the new experiments and clarifications addressed their concerns and called them “valuable additions.” The score was raised from 2 to 6 on **24 Nov 2025**.

- **Reviewer `kjZH`**
  - Rating: 8 (unchanged)

---

### Meta-Review · Area_Chair_CVqt · 2026-01-06

**Summary:**

Several major concerns were raised by the reviewers: 1) Complexity of the algorithm; 2) the ablation study that uses raw video generation model; 3) failure cases; 4) comparing with Chain-of-Thought based methods.

**Reviewer Concerns:**

Reviewers have raised several major concerns. Some of them are addressed by the authors, e.g., 1) the computational complexity report; 2) the ablation study that uses raw video generation model and 3) failure cases.

There are issues that are only partially addressed by the authors: 1) the impact of the base video generation model on the resulting image edting model; 2) the comparison between Chain-of-Thought based methods.

**Reviewer Scores:**

It seems the reviewer-author discussion is complete. The change of scores has already been proposed by the reviewer.

---

### Decision · Program_Chairs · 2026-01-26

Accept (Poster)